# Signed in Ink, Hidden in Noise: Watermarking Diffusion Large Language Models

## Abstract

Watermarking techniques enable the embedding of imperceptible signals into autoregressive large language models (LLMs), facilitating reliable detection and attribution of AI-generated text. However, for the emerging paradigm of diffusion large language models (dLLMs), which provide bidirectional context modeling, greater generation flexibility and controllability, and more efficient sampling, research on watermarking remains largely unexplored, raising critical concerns about copyright protection. In this work, we present the first systematic investigation of watermarking for dLLMs and introduce Ripple, a dedicated framework specifically designed for the diffusion-based generation process. The Ripple operates in two complementary stages, namely watermark injection and watermark calibration, to achieve seamless integration of watermark signals within dLLMs. Furthermore, extensive experiments on two representative dLLMs across multiple datasets demonstrate that Ripple effectively balances detectability, robustness, and text quality. We believe this study provides a foundation for advancing future research on watermarking in dLLMs.

## 1 Introduction

As generative language models increasingly approximate human writing, the concept of authorship faces unprecedented challenges. When any given sentence may originate from a machine, accurately identifying its true source becomes crucial for ensuring accountability, provenance, and trust. To address this issue, watermarking techniques have been developed to embed imperceptible signals into the outputs of autoregressive language models, enabling reliable detection and attribution. These approaches have shown promising results on mainstream autoregressive architectures. However, with the emergence of diffusion-based large language models (Nie et al., 2025b; Cheng et al., 2025; Ye et al., 2025; Labs et al., 2025), which offer smoother, more diverse, and more controllable text generation, the study of watermarking in this new paradigm remains largely unexplored.

Diffusion Large language models (dLLMs) generate text through an iterative denoising process, offering inherent advantages such as bidirectional context modeling, enhanced flexibility and controllability, and theoretically greater efficiency than autoregressive models. However, the same mechanism that underpins these strengths also poses unique challenges for watermarking. In diffusion-based generation, the timing of watermark injection is crucial: signals inserted too early or too late are prone to being overwritten by subsequent denoising steps, which severely hinders reliable detection. Existing watermarking techniques, designed for sequence-level token generation, cannot be directly applied because dLLMs introduce a temporal dimension on top of the sequential structure of text. Consequently, watermarking in dLLMs shifts from a two-dimensional sequence embedding problem into a three-dimensional one that spans both sequence and diffusion steps, demanding new mechanisms that remain simultaneously detectable and resilient under this generative paradigm.

To address this need, we present the first watermarking framework specifically designed for dLLMs. In contrast to traditional watermarking techniques designed for autoregressive LLMs, our approach operates directly within the diffusion process. Rather than injecting signals through abrupt manipulation of token-level outputs, it progressively embeds watermark information that co-evolving with the generative trajectory across denoising steps, as illustrated in Figure 1. This design aligns with the core principle of diffusion models, in which semantic structure gradually emerges from noise, thereby integrating watermark signals in a distributed and generation-aligned manner.

Specifically, we propose **Ripple**, a watermarking framework seamlessly integrated into the iterative generation workflow of dLLMs. Beginning with a fully masked sequence, dLLMs progressively refine text through multiple diffusion steps. Ripple exploits this process through two synergistic stages. In the **watermark injection stage**, we reorder the remasking operation to embed watermark signals into the tokens actively unmasked at each step, allowing the watermark to co-evolve with the emerging sequence. This design prevents the the embedded watermark from dissipating during subsequent diffusion steps and avoids redundant embedding in the same token. In the **watermark calibration stage**, once the final step is reached or all tokens have been generated, a detector as-

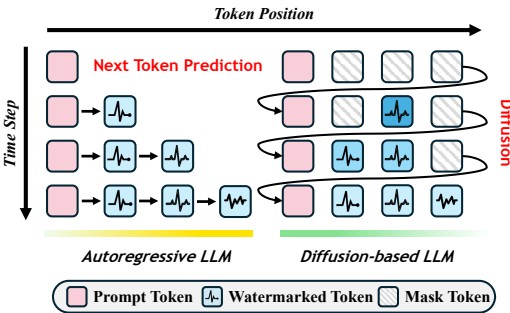

Figure 1: In autoregressive LLMs, tokens are generated sequentially left to right, whereas in diffusion-based LLMs, tokens are progressively denoised through a series of diffusion steps.

sesses the strength of the embedded watermark while a quality-aware evaluator monitors fluency. Tokens that are weakly watermarked or exhibit low quality are corrected through targeted resampling, ensuring that the final output remains both high-quality and strongly traceable. By embedding watermarking into the temporal–sequential dynamics of dLLMs, Ripple transforms watermarking from a static overlay into a generation-aware approach, yielding secure and reliably detectable text in diffusion large language models.

We conduct extensive experiments on mainstream dLLMs as well as a wide range of downstream task benchmarks. The results consistently demonstrate the superiority of the Ripple. Beyond performance, our approach offers a flexible mechanism that dynamically adjusts watermark strength and text fluency across diverse application scenarios, paving the way for practical and trustworthy watermarking solutions in the era of dLLMs. Our key contributions are summarized as follows:

- We introduce Ripple, the first exploration of a dedicated watermarking framework for dLLMs, which harmoniously integrates watermarking into the diffusion-based generation process.

- We carry out comprehensive evaluations across diverse datasets and model architectures, showing that Ripple consistently delivers superior performance under various challenging conditions.

## 2 PRELIMINARIES

### 2.1 DIFFUSION LARGE LANGUAGE MODELS (DLLMS)

**Notations.** Given a dLLM $\Theta$, let $\mathcal{V}$ denote the vocabulary, $|\mathcal{V}|$ denote the vocabulary size, $L$ denote the sequence length, $T$ denote the total diffusion steps, and $p_0$ denote the input prompt. We use $\mathcal{M}_t$ and $\mathcal{U}_t$ to represent the sets of masked and unmasked positions at diffusion step $t$, respectively. The generated sequence at diffusion step $t$ is denoted by $\mathbf{y}_t = (y_t^1, \cdots, y_t^L)$.

**Inference.** During inference, at $t = 0$, the unmasked set is empty ($\mathcal{U}_0 = \varnothing$) and the masked set covers all positions ($\mathcal{M}_0 = \{1, \cdots, L\}$). The dLLM gradually denoises an initial fully masked sequence $\mathbf{y}_0$ and generates $L/T$ new tokens at each diffusion step until generating a completely unmasked sequence $\mathbf{y}_T$. At an intermediate step $t$, the dLLM first takes the prompt $p_0$ together with the sequence $\mathbf{y}_{t-1}$ from the previous $t-1$ steps as input to predict tokens for all positions in $\mathcal{M}_{t-1}$. Subsequently, dLLMs apply a predefined remask strategy (see Appendix L) to update the masked and unmasked sets: $L/T$ tokens in $\mathcal{M}_{t-1}$ is removed to $\mathcal{U}_{t-1}$, forming $\mathcal{M}_t$ and $\mathcal{U}_t$. Next, the tokens in $\mathcal{M}_t$ are remasked, while the tokens in $\mathcal{U}_t$ are retained, producing the sequence $\mathbf{y}_t$ at step $t$. This iterative process continues, with the number of masked tokens gradually decreasing as the diffusion steps proceed, until either $t = T$ or $|\mathcal{M}_t| = 0$, yielding the final sequence $\mathbf{y}_T$.

### 2.2 AUTOREGRESSIVE LLM WATERMARK

**Notations.** With an autoregressive LLM, given a prompt $p_0$ and a sequence of prior tokens $\mathbf{x}_{<i} = (x_1, \cdots, x_{i-1})$, the LLM computes the next token probability distribution $p_M(\cdot \mid p_0, \mathbf{x}_{<i})$ and

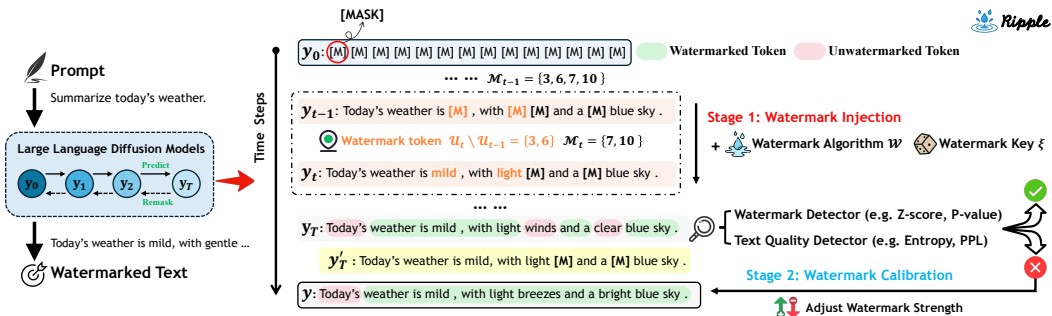

Figure 2: A Versatile Watermarking Framework Ripple for Diffusion Large Language Models.

samples the next token $x_i$ in the vocabulary $\mathcal{V}$. LLM watermarking $\mathcal{W}$ for token generation can be categorized into **(i) logits-based algorithm** $\mathcal{W}_l$ and **(ii) sampling-based algorithm** $\mathcal{W}_s$.

**(i) Logits-based** algorithms embed watermark signals by modifying the logits distribution. A pioneering approach is KGW (Kirchenbauer et al., 2023), which partitions the vocabulary into red and green tokens according to a proportion parameter $\gamma$. Then the preceding $k$ tokens are hashed using a watermark key $\xi$, and a bias $\delta$ is added to the logits of green tokens to increase their selected probability. During detection, hypothesis testing is performed on the number of green tokens $n_{\mathrm{G}}$ in a text of length $L$. If the $z$-score $= (n_{\mathrm{G}} - \gamma L)/\sqrt{L\gamma(1-\gamma)}$ surpasses a predefined threshold, indicating that the proportion of green tokens significantly exceeds $\gamma$, the text is considered watermarked.

**(ii) Sampling-based** algorithms embed watermarks by modifying the original sampling method $O$, without changing the logits. A representative work is EXP (Aaronson, 2023), which uses an exponential minimum sampling scheme to select the token $x_i$ that satisfies $\arg\max_{v \in \mathcal{V}} (r_v^i)^{1/p_v^i}$ at position $i$, where the LLM outputs a probability vector $\mathbf{p}^i = (p_1^i, \cdots, p_{|\mathcal{V}|}^i)$, and a pseudo-random vector $\mathbf{r}^i = (r_1^i, \cdots, r_{|\mathcal{V}|}^i)$ is obtained by a predefined watermark key. During detection, the correlation score between the generated text and the pseudo-random vector sequence is measured to determine whether it exceeds a set threshold, which would signal the presence of a watermark.

## 3 METHOD

This section defines watermarking for dLLMs, analyzes the challenges it faces, and introduces a watermarking framework, Ripple, for dLLMs, as illustrated in Figure 2. Ripple comprises three key components: progressively diffused watermark injection, detectability-quality harmonized calibration, and watermark detection.

### 3.1 TASK FORMALIZATION

**Definition 3.1** (dLLM watermarking). A large language diffusion model watermarking scheme consists of two probabilistic polynomial-time algorithms (Watermark, Detect):

$$\mathbf{y} = \mathsf{Watermark}(\Theta, p_0, \xi), \ s = \mathsf{Detect}(\mathbf{y}) \tag{1}$$

where $\xi$ denotes watermark key, with $s = 1$ for watermarked text and $s = 0$ otherwise.

**Challenge.** Since dLLMs compute logits and sample tokens for all positions at each diffusion step, directly applying existing watermarking methods (both logits-based and sampling-based) to dLLMs face two key challenges. First, after watermark tokens are generated at each step, a large portion of them are remasked, causing the watermark signal to dissipate. Second, applying a watermarking algorithm to all token positions at every step would severely degrade generation efficiency and harm text quality. To address the above challenges, we propose the Ripple framework as follows.

---

**Algorithm 1:** Diffusion Large Language Models Watermarking (RIPPLE)

---

**Input:** Target model $\Theta$, prompt $p_0$, diffusion steps $T$, length $L$, watermark key $\xi$, watermark algorithm $\mathcal{W}$
**Output:** Watermarked Text $\mathbf{y}$

1 **procedure** RIPPLE:
    // Stage 1: Watermark Injection
2     Initialize an all-mask sequence of length L: $\mathbf{y}_0 \leftarrow [\mathbf{M}] * L$ // [M] denotes mask token
3     Initialize masked index and unmasked set: $\mathcal{M}_0 \leftarrow \{1, \cdots, L\}, \mathcal{U}_0 \leftarrow \varnothing$
4     **for** $t \leftarrow 1, \cdots, T$ **do**
5         Feed the prompt and the $t-1$ step output into the model to get logits: $\boldsymbol{L}_t = \Theta(p_0, \mathbf{y}_{t-1})$
6         Apply softmax to the logits to obtain the probability matrix: $\boldsymbol{P}_t = \mathsf{softmax}(\boldsymbol{L}_t)$
7         Locate current remask and unmask positions set: $\mathcal{U}_t, \mathcal{M}_t \leftarrow \mathsf{Remask}(\mathcal{M}_{t-1}, \boldsymbol{P_t})$
8         Embed watermark at added positions: $y_t^{\mathcal{U}_t \setminus \mathcal{U}_{t-1}} \leftarrow \mathsf{Watermark}(\Theta, \mathbf{y}_{t-1}, \xi)$
9         Mask tokens in the remask positions set: $y_t^{\mathcal{M}_t} \leftarrow [\mathbf{M}]$
        // Stage 2: Watermark Calibration
10         **if** $t == T$ or $|\mathcal{M}_t| = 0$ **then**
11             Compute score $z_1$ with watermark detector: $z_1 = \mathsf{Detect}(\Theta, \mathbf{y}_T, \xi)$
12             Compute score $z_2$ with perplexity evaluator: $z_2 = \mathsf{PPL}(\Theta, \mathbf{y}_T)$
13             **if** $z_1 < \alpha$ or $z_2 > \beta$ **then**
14                 If the threshold is met, apply correction: $\mathbf{y} \leftarrow \mathsf{Calibration}(\Theta, \mathbf{y}_T, \xi)$
15             **end**
16             Output the final sequence: **return** $\mathbf{y}$
17         **end**
18     **end**
19 **end**

---

### 3.2 PROGRESSIVELY DIFFUSED WATERMARK INJECTION

In the Ripple framework, we move the remask operation before generating tokens for all positions in dLLMs. At each diffusion step, immediately after generating the logits matrix, we identify the unmask positions based on a predefined remask strategy (see Appendix L). The watermark is then embedded only at these positions, anchoring them to stable locations throughout the diffusion process. This progressive embedding preserves watermark integrity while reducing redundant computations.

Specifically, as shown in lines **2-9** of Algorithm 1, the process begins with a fully masked sequence $\mathbf{y}_0$. At each diffusion step $t$, the model $\Theta$ takes the prompt $p_0$ and previous sequence $\mathbf{y}_{t-1}$ to produce logits matrix $\boldsymbol{L}_t = (\boldsymbol{l}_t^1, \cdots, \boldsymbol{l}_t^L) \in \mathbb{R}^{L \times |\mathcal{V}|}$, which is then converted to a token probability matrix $\boldsymbol{P}_t = (\boldsymbol{p}_t^1, \cdots, \boldsymbol{p}_t^L)$ via $\mathsf{softmax}(\cdot)$. Using $\boldsymbol{P}_t$ and previous masked set $\mathcal{M}_{t-1}$, the remasking strategy $\mathsf{Remask}(\cdot)$ determines the masked set $\mathcal{M}_t$ and the unmasked set $\mathcal{U}_t$. Next, to prevent the injected watermark signal from being masked in later steps or redundantly reinjected, we embed the watermark only into the newly generated tokens at each diffusion step, specifically those in $\mathcal{U}_t \setminus \mathcal{U}_{t-1}$ to obtain $y_t^{\mathcal{U}_t \setminus \mathcal{U}_{t-1}}$ as Equation 2, meanwhile we apply remask to the positions in $\mathcal{M}_t$ to get $y_t^{\mathcal{M}_t}$.

$$y_t^{\mathcal{U}_t \setminus \mathcal{U}_{t-1}} = \begin{cases} O(\mathsf{softmax}(\mathcal{W}_l(\boldsymbol{L}_t, \xi))) \\ \mathcal{W}_s(\mathsf{softmax}(\boldsymbol{L}_t), \xi) \end{cases} \tag{2}$$

Crucially, during watermark injection, we employ a global hash scheme, ensuring that the hash computations remain context-independent across diffusion steps. Other position-dependent hashing schemes are difficult to apply, as the exact generation order cannot be reliably reconstructed during detection. This design mitigates positional uncertainty and allows for a smooth adaptation of watermarking techniques originally developed for LLMs. Furthermore, in adversarial scenarios, the use of a global hash scheme significantly enhances watermark robustness, and subsequent experiments show that it has minimal impact on text quality. More relevant discussions and theoretical conclusions can be found in the Appendix M.

### 3.3 DETECTABILITY-QUALITY HARMONIZED CALIBRATION

Compared with autoregressive LLMs, dLLMs offer an inherent advantage: they can modify tokens at arbitrary positions simply by adding extra diffusion steps, avoiding left-to-right regeneration.

Inspired by this, we further design a watermark calibration mechanism that selectively refines tokens, enabling a better balance between detectability and text quality (see Lines **10–16** of Algorithm 1).

**Enhancing Detectability.** Given a generated sequence $\mathbf{y}_T$, we first apply the designated watermark detector $\text{Detect}(\cdot)$ to compute its watermark score $z_1$. If $z_1$ exceeds the watermark threshold $\alpha$, the sequence is regarded as successfully watermarked. Otherwise, we remask the tokens that contribute minimally to the watermark signal, which are assigned low confidence by the detector, for regeneration with stronger watermark strength. During this process, tokens with higher entropy are prioritized, as modifying low-entropy tokens tends to have a greater impact on text quality:

$$H^i = -\sum_{j \in \mathcal{V}} \mathbf{p}_j^i \log \mathbf{p}_j^i, \;\; i \in \{1, \cdots, L\} \tag{3}$$

**Improving Text Quality.** Additionally, we assess the fluency of each generated sequence $\mathbf{y}_T$ using the perplexity evaluator $\text{PPL}(\cdot)$. If the PPL value $z_2$ of $\mathbf{y}_T$ is below the perplexity threshold $\beta$, the sequence is considered fluent and requires no further modification. Otherwise, if $z_2$ exceeds $\beta$, we calculate the self-information of each token $y_T^i$ and remask those with high self-information:

$$I_T^i = -\log p_\Theta(y_T^i \mid y_T^1, \cdots, y_T^{i-1}, y_T^{i+1}, \cdots, y_T^L), \; i \in \{1, \cdots, L\} \tag{4}$$

Higher self-information indicates that the token contributes more to the overall perplexity of the sentence. Consequently, tokens with higher self-information are selectively remasked and regenerated with a lower watermark strength. Notably, the remasking ratio $\eta$ is a dynamically adjustable hyperparameter, allowing for a flexible trade-off between watermark detectability and text quality.

## 3.4 DLLMS WATERMARK DETECTION

During watermark detection, our approach is model-agnostic, allowing detection algorithms developed for LLMs to be directly applied to dLLMs with a global hash scheme. Specifically, we first select a watermark detector consistent with the employed watermarking scheme and use the watermark key $\xi$ as input to compute a watermark score for each generated text. We then perform hypothesis testing by evaluating the corresponding statistics against a predefined threshold: texts exceeding the threshold are classified as watermarked, whereas those below are considered unwatermarked. In practice, a sliding-window mechanism can be adopted to conduct detection on text segments. Detailed formulations of the detection functions are provided in Appendix E.

## 4 EXPERIMENTAL SETUP

**Datasets.** To evaluate the detectability and fluency of watermarking methods on dLLMs, we follow Kirchenbauer et al. (2023); Zhao et al. (2024) and conduct experiments on subsets of the C4 (Raffel et al., 2020) and OpenGen (Krishna et al., 2023) datasets, randomly sampling 200 examples per run. For each prompt, we generate both watermarked and non-watermarked outputs, each comprising 128 tokens. To measure the impact on text quality, we use the WaterBench (Tu et al., 2024) benchmark across four downstream tasks. More dataset details are provided in Appendix F.

**Metrics.** We evaluate watermark detectability using the True Positive Rate (TPR), True Negative Rate (TNR), False Positive Rate (FPR), and F1 Score, while text fluency is assessed using Perplexity (PPL) and GPT-4 Score. To evaluate downstream task performance, we use generation metrics (GM) such as ROUGE-L and Edit Similarity. Additionally, the robustness of watermarking methods is measured using the AUROC curve. Detailed definitions are provided in Appendix G.

**Models.** We use two mainstream dLLMs: the Dream series (Dream-7B-Base/Instruct) (Ye et al., 2025) and the LLaDA series (LLaDA-8B-Base/Instruct) (Nie et al., 2025b), as detailed in Appendix C. For comparison, we include the autoregressive LLM LLaMA3-8B-Instruct (Dubey et al., 2024), and LLaMA2-13B-Chat-hf (Touvron et al., 2023) is used for PPL evaluation to ensure fairness.

| Watermark | Version | C4 DATASET | | | | | | | | OPENGEN DATASET | | | | | | | |
| | | DREAM-7B-BASE | | | | LLaDA-8B-BASE | | | | DREAM-7B-BASE | | | | LLaDA-8B-BASE | | | |
| | | TPR ↑ | TNR ↑ | F1 ↑ | PPL ↓ | TPR ↑ | TNR ↑ | F1 ↑ | PPL ↓ | TPR ↑ | TNR ↑ | F1 ↑ | PPL ↓ | TPR ↑ | TNR ↑ | F1 ↑ | PPL ↓ |
| None | - | - | - | - | 3.913 | - | - | - | 3.603 | - | - | - | 4.200 | - | - | - | 3.710 |
| Unigram | **Van.** | 0.955 | 0.910 | 0.934 | 7.934 | 0.950 | 0.935 | 0.943 | 6.693 | 0.935 | 0.885 | 0.912 | 8.201 | 0.905 | 0.925 | 0.914 | 6.870 |
| | **Rip.** | 0.995 | 0.935 | **0.966** | 5.323 | 0.940 | 0.950 | **0.945** | 5.617 | 0.990 | 0.965 | **0.978** | 5.769 | 0.985 | 0.975 | **0.980** | 5.875 |
| EXP | **Van.** | 0.955 | 0.985 | 0.970 | 8.691 | 0.965 | 0.990 | 0.977 | 7.920 | 0.990 | 0.935 | 0.964 | 9.705 | 0.970 | 0.990 | 0.980 | 7.985 |
| | **Rip.** | 0.965 | 0.980 | **0.972** | 5.734 | 0.990 | 0.995 | **0.992** | 5.802 | 0.940 | 1.000 | **0.969** | 6.023 | 0.990 | 0.990 | **0.990** | 5.381 |
| SWEET | **Van.** | 0.880 | 0.910 | 0.893 | 7.129 | 0.855 | 0.725 | 0.803 | 6.708 | 0.895 | 0.945 | 0.918 | 7.910 | 0.920 | 0.590 | 0.790 | 7.320 |
| | **Rip.** | 0.905 | 0.965 | **0.933** | 6.485 | 0.880 | 0.800 | **0.846** | 6.033 | 0.945 | 0.960 | **0.952** | 7.022 | 0.840 | 0.890 | **0.862** | 7.268 |
| EWD | **Van.** | 0.945 | 0.890 | 0.920 | 7.456 | 0.960 | 0.920 | 0.941 | 7.009 | 0.985 | 0.930 | 0.959 | 8.158 | 0.940 | 0.905 | 0.924 | 7.202 |
| | **Rip.** | 0.960 | 0.950 | **0.955** | 4.833 | 0.935 | 0.950 | **0.942** | 5.575 | 0.985 | 0.985 | **0.985** | 5.459 | 0.965 | 0.950 | **0.958** | 6.429 |
| UPV | **Van.** | 0.910 | 0.910 | 0.910 | 6.729 | 0.935 | 0.865 | 0.903 | 7.776 | 0.870 | 0.755 | 0.823 | 7.714 | 0.895 | 0.890 | 0.893 | 7.351 |
| | **Rip.** | 0.965 | 0.985 | **0.975** | 5.295 | 0.965 | 0.945 | **0.955** | 6.942 | 0.975 | 0.945 | **0.961** | 7.145 | 0.980 | 0.970 | **0.975** | 6.587 |
| MorphMark | **Van.** | 0.900 | 0.880 | 0.891 | 6.896 | 0.915 | 0.880 | 0.899 | 6.302 | 0.945 | 0.890 | 0.920 | 7.301 | 0.925 | 0.830 | 0.883 | 6.436 |
| | **Rip.** | 0.925 | 0.880 | **0.905** | 6.457 | 0.880 | 0.955 | **0.914** | 4.923 | 0.945 | 0.935 | **0.940** | 6.937 | 0.920 | 0.935 | **0.927** | 5.680 |
| SynthID | **Van.** | 0.980 | 0.960 | 0.970 | 5.489 | 0.945 | 0.975 | 0.959 | 5.605 | 0.945 | 0.955 | 0.950 | 5.885 | 0.935 | 0.985 | 0.959 | 5.155 |
| | **Rip.** | 0.965 | 0.980 | **0.972** | 5.411 | 0.955 | 0.975 | **0.965** | 5.322 | 0.955 | 0.965 | **0.960** | 5.740 | 0.945 | 0.980 | **0.962** | 5.100 |

Table 1: The detectability and PPL of Dream-7B and LLaDA-8B under different watermarking algorithms on C4 and OpenGen. **Rip.** denotes our complete framework Ripple, while **Van.** refers to vanilla performance without watermark calibration stage, serving as an ablation comparison study.

**Baselines.** We evaluate multiple watermarking methods from LLMs to dLLMs, including logits-based approaches (Unigram (Zhao et al., 2024), SWEET (Lee et al., 2024), EWD (Lu et al., 2024), UPV (Liu et al., 2024a), MorphMark (Wang et al., 2025b)) and sampling-based methods (EXP (Aaronson, 2023), SynthID (Dathathri et al., 2024)). Detailed baseline settings are in Appendix H.

**Implementation Details.** Our method is implemented using Python 3.10 and PyTorch 2.6, and all experiments are conducted on two NVIDIA A100 80G GPUs. For different hyperparameters, we set the generation sequence length to $L = 128$, the number of diffusion steps to $T = 128$, the sampling parameters to top-$k = 64$ and top-$p = 0.95$, the watermark detection threshold to $\alpha = 4.0$, the perplexity threshold to $\beta = 10.0$, the remask strategy to entropy-based and the remask ratio to $\eta = 0.25$. Hyperparameter analysis is provided in Section 5.4.

## 5 ANALYSIS

To validate the feasibility of watermarking in dLLMs, we conduct a comprehensive evaluation from three perspectives: detectability, text quality, and robustness. Experimental results demonstrate that our Ripple framework perfectly integrates the watermarking into the dLLMs, achieving strong detectability and robustness while maintaining high-quality text generation.

### 5.1 WATERMARK DETECTABILITY

Table 1 provides a comprehensive evaluation of various watermarking schemes adapted for dLLMs, tested across mainstream dLLMs and two datasets. The results demonstrate that our Ripple framework enables the seamless migration of watermarking algorithms to dLLMs, consistently achieving an average F1 score above **0.95** and a TPR close to **1.0**, making its watermarking performance comparable to that of autoregressive LLMs.

**Impact of watermark calibration.** Furthermore, compared to vanilla watermarking approaches without the watermark calibration module, the complete Ripple framework further enhances watermark detectability without compromising text fluency. Specifically, for watermarking methods with initially low detection rates (e.g., SWEET and UPV), Ripple improves the average F1 score by **4.73%** and **8.43%**, respectively. Even for high-performing methods such as EWD and Unigram, Ripple still yields F1 gains of **2.4%** and **4.15%**, respectively. These findings underscore the advantage of dLLMs in enabling the correction of selected tokens without regenerating the entire text. They also demonstrate the effectiveness of integrating both watermark and text quality evaluators into additional diffusion steps, highlighting the flawless fusion of watermarking with the diffusion process enabled by the Ripple framework.

| Model | T1: Short Q, Short A _Factual Knowledge_ | | | | T2: Short Q, Long A _Long-form QA_ | | | | T3: Long Q, Short A _Math Reasoning_ | | | | T4: Long Q, Long A _Summarization_ | | | |
|---|---|---|---|---|---|---|---|---|---|---|---|---|---|---|---|---|
| + Watermark | FPR↓ | F1↑ | GM↑ | DROP | FPR↓ | F1↑ | GM↑ | DROP | FPR↓ | F1↑ | GM↑ | DROP | FPR↓ | F1↑ | GM↑ | DROP |
| LLAMA3-8B-INSTRUCT | - | - | 56.50 | - | - | - | 23.09 | - | - | - | 37.94 | - | - | - | 37.45 | - |
| + Unigram | **0.930** | 0.674 | 48.50 | ↓14.2% | 0.070 | 0.935 | 22.36 | ↓3.16% | 0.240 | 0.822 | 29.48 | ↓22.3% | 0.210 | 0.822 | 35.13 | ↓6.19% |
| + EXP | 0.975 | 0.667 | 48.50 | ↓14.2% | **0.035** | 0.944 | 22.67 | ↓1.82% | 0.670 | 0.788 | 33.90 | ↓10.6% | 0.280 | 0.781 | 21.22 | ↓43.3% |
| + EWD | 0.981 | 0.673 | **55.00** | ↓2.65% | 0.055 | 0.942 | 22.02 | ↓4.63% | 0.440 | 0.790 | **36.98** | ↓2.53% | 0.245 | 0.814 | 35.49 | ↓5.23% |
| + UPV | 0.995 | 0.668 | 50.50 | ↓10.6% | 0.065 | 0.943 | 22.68 | ↓1.78% | **0.227** | 0.796 | 29.88 | ↓21.2% | 0.130 | 0.873 | 34.74 | ↓7.34% |
| + SynthID | 1.000 | 0.667 | 54.00 | ↓4.42% | 0.070 | 0.933 | **22.71** | ↓1.65% | 0.265 | 0.716 | 35.34 | ↓6.85% | **0.100** | 0.851 | **36.19** | ↓3.36% |
| DREAM-7B-INSTRUCT | - | - | 63.50 | - | - | - | 19.05 | - | - | - | 38.46 | - | - | - | 39.63 | - |
| + Unigram | 0.995 | 0.666 | 63.00 | ↓0.79% | 0.075 | 0.897 | 17.00 | ↓10.8% | **0.290** | 0.729 | 35.62 | ↓7.35% | 0.145 | 0.820 | 34.29 | ↓13.5% |
| + EXP | 0.965 | 0.672 | 60.00 | ↓5.51% | 0.005 | 0.964 | 13.61 | ↓28.6% | 0.940 | 0.678 | 34.11 | ↓11.3% | **0.010** | 0.926 | 26.69 | ↓32.7% |
| + EWD | **0.635** | 0.700 | 62.50 | ↓1.57% | 0.000 | 0.997 | **17.85** | ↓6.30% | 0.370 | 0.790 | 32.75 | ↓14.8% | 0.050 | 0.963 | **36.38** | ↓8.20% |
| + UPV | 0.985 | 0.666 | 62.00 | ↓2.36% | **0.000** | 1.000 | 14.37 | ↓24.6% | 0.885 | 0.677 | 27.82 | ↓27.7% | 0.080 | 0.878 | 35.27 | ↓11.0% |
| + SynthID | 0.995 | 0.666 | **63.50** | ↓0.00% | 0.066 | 0.927 | 17.56 | ↓7.82% | 0.845 | 0.675 | **36.40** | ↓5.36% | 0.105 | 0.875 | 34.23 | ↓13.6% |
| LLADA-8B-INSTRUCT | - | - | 60.50 | - | - | - | 23.41 | - | - | - | 39.46 | - | - | - | 45.39 | - |
| + Unigram | 0.980 | 0.669 | 57.00 | ↓5.79% | 0.005 | 0.998 | 21.07 | ↓10.0% | 0.235 | 0.874 | 32.64 | ↓17.3% | 0.205 | 0.830 | **38.09** | ↓16.1% |
| + EXP | 1.000 | 0.667 | 60.00 | ↓0.83% | 0.025 | 0.988 | 19.53 | ↓16.6% | 0.995 | 0.668 | 34.76 | ↓11.9% | **0.060** | 0.910 | 20.14 | ↓55.6% |
| + EWD | **0.440** | 0.688 | **60.50** | ↓0.00% | **0.000** | 0.995 | 21.84 | ↓6.71% | **0.225** | 0.769 | 33.46 | ↓15.2% | 0.115 | 0.882 | 28.77 | ↓36.6% |
| + UPV | 1.000 | 0.664 | 56.00 | ↓7.44% | 0.055 | 0.934 | 21.36 | ↓8.76% | 0.655 | 0.681 | 35.21 | ↓10.8% | 0.165 | 0.802 | 30.38 | ↓33.1% |
| + SynthID | 1.000 | 0.667 | 57.50 | ↓4.96% | 0.035 | 0.957 | **22.68** | ↓3.12% | 0.570 | 0.694 | **39.11** | ↓0.89% | 0.180 | 0.808 | 36.96 | ↓18.6% |

Table 2: The performance of various watermarking algorithms across four different downstream tasks on LLM and dLLMs, using FPR, F1, GM and , and Generation Quality Drop (Drop).

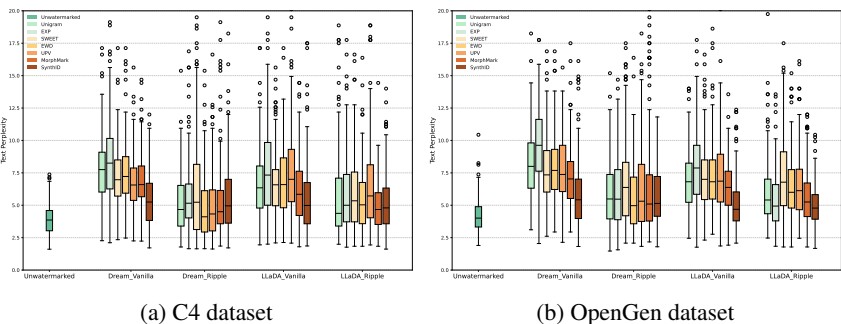

(a) C4 dataset        (b) OpenGen dataset

Figure 3: A comparison of PPL across various dLLMs watermarking with Ripple Framework.

## 5.2 TEXT QUALITY

**Perplexity (PPL).** We evaluate the fluency of watermarked text generated by dLLMs using the PPL metric. As illustrated in Table 1 and Figure 3, the Ripple framework substantially reduces the PPL of watermarked text compared to the vanilla version, achieving fluency nearly on par with non-watermarked outputs, with only a slight increase. This demonstrates that our watermarking framework has a relatively minor impact on the fluency of original texts.

**Downstream Task.** To evaluate the quality of dLLM-generated watermarked text, we test the Ripple framework on four downstream tasks of varying input and output lengths (Table 2) and compare it with LLaMA3-8B-Instruct using the original watermarking method. Results show that Ripple consistently achieves high detection rates while maintaining strong text quality, especially on longer-output tasks. Remarkably, on certain tasks, dLLMs even outperform the advanced LLaMA3 model. For example, by applying the Ripple framework, Dream achieves a SynthID watermarking score that surpasses LLaMA3 by **1.06** points on Task 3, while LLaDA attains an Unigram watermarking score exceeding LLaMA3 by **2.96** points on Task 4. These results further validate the potential and feasibility of watermarking in dLLMs. The underlying intuition is that Ripple inherits the dLLMs' ability to leverage bidirectional context and perform inference in an arbitrary token order, enabling more effective handling of multi-constraint tasks and the pursuit of specific generation objectives.

## 5.3 ROBUSTNESS TO REAL-WORLD ATTACKS

To ensure that our watermarking scheme remains resilient against a wide range of real-world attacks (Kirchenbauer et al., 2024), we conduct a comprehensive robustness evaluation of various watermarking methods on dLLMs, including edit attacks, copy-paste attacks, back-translation, and

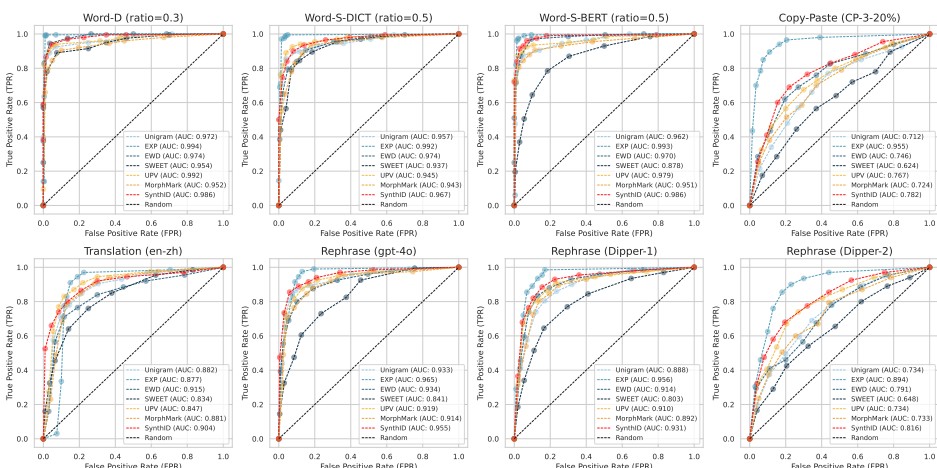

Figure 4: The AUROC curve of LLaDA-8B-Base watermarked text under various attacks on C4.

paraphrasing attacks performed using models such as GPT-4o and Dipper (Krishna et al., 2023). Detailed attack settings are provided in Appendix I.

As illustrated by the ROC curves and AUC scores in Figure 4, different watermarking algorithms exhibit varying levels of robustness on LLaDA under all attack scenarios. Among them, the EXP watermarking algorithm based on the Ripple framework achieves the highest robustness, with an average AUC of **0.953**, thanks to its exponential minimal sampling mechanism. Other watermarking methods also perform well overall, though their effectiveness decreases when confronted with stronger attacks such as those from Dipper and copy-paste operations. For instance, Unigram and SynthID achieve average AUC of **0.880** and **0.916**, respectively. This resilience is largely attributed to the use of a global hash scheme during watermark injection, which ensures that the watermark signals of individual tokens are independent, allowing detection even when token positions are altered by attacks. Additional attack results can be found in Appendix B.

## 5.4 FURTHER ANALYSIS

**Impact of remask strategy.** As shown in Figure 5, we evaluate the performance of the LLaDA-8B-Base model under various remask strategies, including random, margin-based, entropy-based, and confidence-based approaches (see Appendix L for details). We report both the watermark F1 score and the GPT-4 score, the latter reflecting the text quality as judged by GPT-4 (template in Appendix J). The results show that, except for the random baseline, all other strategies exhibit relatively stable performance, with minimal fluctuations in both F1 and GPT-4 scores. Ultimately, we adopt the entropy-based remask strategy as the default, since entropy captures token-level uncertainty: prioritizing low-entropy tokens helps preserve text quality, while high-entropy tokens are more favorable for watermark injection.

**Hyperparameter analysis.** We systematically investigate the impact of key hyperparameters on watermark performance and text quality (see appendix D for more details):

- **Diffusion Steps.** As shown in Figure 6a, increasing the number of diffusion steps $T$ from 1 to 128 steadily improves the watermark F1 score and reduces perplexity due to enchanced denoising. Therefore, we set $T$ equal to the sequence length $L$, generating one token per step.

- **Watermark Strength.** As shown in Figure 6b, stronger signals reduce the FPR but harm fluency. To balance this trade-off, we initialize $\delta = 2.0$. During the watermark calibration stage, we increase it to $4.0$ if the watermark score $z_1$ falls below the threshold $\alpha$, and decrease it to $1.5$ if the perplexity score $z_2$ exceeds the fluency threshold $\beta$.

- **Remask Ratio.** As shown in Figure 6c, varying the remask ratio has only minor effects because we preferentially remask high-entropy tokens, which dominate detectability. Therefore, to achieve a good balance between detectability and text quality, we fix the remask ratio at $\eta = 0.25$.

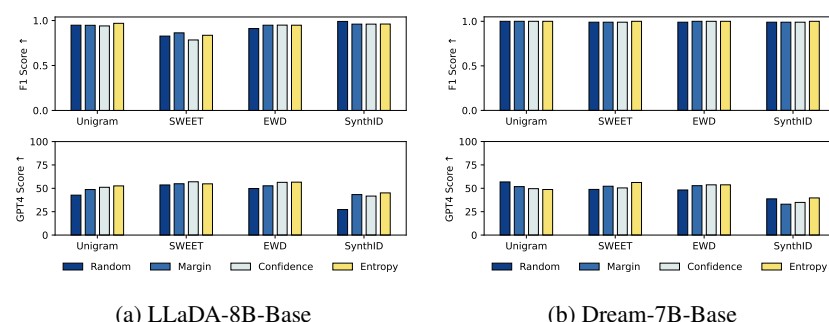

|  |  |
|---|---|
| (a) LLaDA-8B-Base | (b) Dream-7B-Base |

Figure 5: The comparison of different remasking strategies across various watermark algorithms.

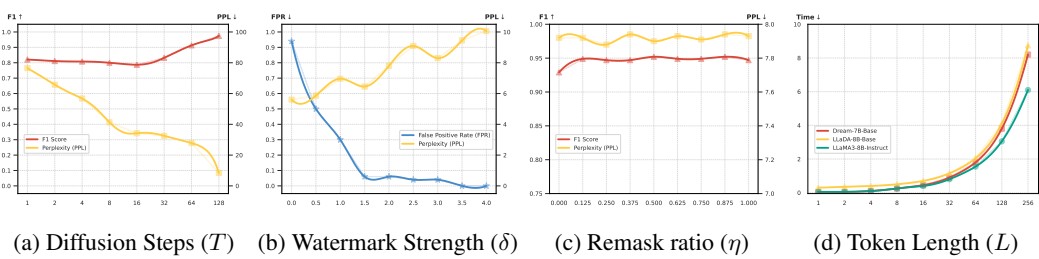

(a) Diffusion Steps ($T$)    (b) Watermark Strength ($\delta$)    (c) Remask ratio ($\eta$)    (d) Token Length ($L$)

Figure 6: Analysis of results with varying hyperparameters.

- **Efficiency Analysis.** As shown in Figure 6d, when each diffusion step produces only one token, generating a 256-token watermarked sequence takes 8.7s for LLaDA, 8.2s for Dream, compared with 6.1s for LLaMA3. While calibration slightly lowers efficiency, it enhances watermark detectability, and future dLLMs may alleviate this overhead by generating multiple tokens per step.

## 6 RELATED WORK

Currently, the most relevant work to ours is the only study targeting Order-agnostic LMs Watermark, PatternMark (Chen et al., 2025). Our work differs from it in: **(1) Design Motivation.** PatternMark primarily focuses on modifying the hash scheme of KGW (Kirchenbauer et al., 2023), proposing a Markov chain–based key sequence generation method and a pattern–based watermark detection technique. In contrast, our Ripple framework mainly focuses on transferring existing watermarking methods from LLMs to dLLMs. **(2) Models and Tasks.** PatternMark mainly targets earlier models (pre-2022) such as LM Protein-MPNN and CMLM, focusing on relatively specific tasks like machine translation and protein synthesis. In contrast, our method is designed for modern dLLMs and supports a wider range of downstream tasks, making it both more practical for current applications.

Notably, we reproduce PatternMark based on its original description and compared it with Ripple under the Unigram watermark. As shown in Table 3 in Appendix B, the results clearly demonstrate that our method achieves higher detectability and robustness across various attack scenarios. Additionally, more related work can be found in the Appendix A.

## 7 CONCLUSION

In this paper, we introduce Ripple, the first framework for watermarking in Diffusion Large Language Models (dLLMs). Ripple comprises two core components: watermark injection and watermark calibration. Extensive experiments show its effectiveness in balancing detectability, robustness, and text quality. In future work, we plan to further explore watermarking in dLLMs from both theoretical and practical perspectives, leveraging their unique advantages. We anticipate that our study will serve as a solid foundation for subsequent research on dLLM watermarking and encourage broader engagement toward secure and responsible use of diffusion-based LLMs.

## 8 ETHICAL STATEMENT

This research focuses on watermarking techniques for diffusion large language models (dLLMs). It does not involve human subjects, personally identifiable information, or sensitive data, and thus does not require IRB approval. All datasets used are publicly available and appropriately licensed. The work aims to advance responsible and transparent watermarking methods for dLLMs and does not introduce harmful content, discriminatory practices, or security vulnerabilities. Large language models were used solely for language polishing and writing assistance in preparing this manuscript; all ideas, methods, experiments, and analyses are original to the authors. No conflicts of interest or external sponsorship influencing the reported findings exist.

## 9 REPRODUCIBILITY STATEMENT

We have made every effort to ensure the reproducibility of our work. Section 3 of the main text and Appendix E provide detailed descriptions of the proposed dLLMs watermarking framework, including both the watermark injection and detection procedures, with the corresponding algorithmic workflows presented in Algorithms 1 and 2. Section 4 and the associated appendices offer further experimental details, case studies, and proofs of our theoretical claims. For example, Appendix F describes the datasets used, Appendix G details the evaluation metrics, Appendix H lists the hyperparameter settings for different watermarking algorithms, and Appendix I specifies the configurations of various attacks. Finally, we include an anonymous supplementary repository containing our source code and experimental scripts to facilitate replication and further research.

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

CONTENTS

## A  MORE RELATED WORK

### A.1  DIFFUSION MODELS IN NLP

Diffusion models (Sohl-Dickstein et al., 2015; Ho et al., 2020) were first designed for image generation and have achieved remarkable success. Yet their adoption in NLP is limited by text's discrete nature (Zou et al., 2023). To address this, two main approaches have emerged. One maps text into a continuous embedding space and applies Gaussian noise (Li et al., 2022; Gong et al., 2023b;a), but often suffers from low efficiency and suboptimal quality. The other works directly in the discrete space via categorical distributions (He et al., 2023; Ou et al., 2024; Nie et al., 2025a; Gong et al., 2025; Labs et al., 2025), progressively corrupting and denoising tokens. Recent models like LLaDA (Nie et al., 2025b) and Dream (Ye et al., 2025) challenge the autoregressive paradigm, showing that non-autoregressive masked diffusion can match or outperform models of similar scale.

### A.2  LLM WATERMARK IN TEXT GENERATION

Current popular LLM watermarking methods fall into two types: logits-based and sampling-based:

**Logits-based Watermarking.** Logits-based watermarking originated with KGW (Kirchenbauer et al., 2023), which randomly splits the vocabulary into "green" and "red" tokens using a hash-based secret key, and boosts the likelihood of green tokens during generation. To improve robustness, Unigram (Zhao et al., 2024) uses a global hash scheme for partitioning. To enhance text quality, methods like Ren et al. (2024); He et al. (2024); Liu et al. (2024b); Liu & Bu (2024); Huo et al. (2024); Chen et al. (2024); WONG et al. (2025) guide the partition using semantic embeddings, while Hu et al. (2024); Wu et al. (2023a) propose unbiased watermarking. Entropy-based improvements are explored by SWEET (Lee et al., 2024), EWD (Lu et al., 2024), and SymMark (Wang et al., 2025a).

**Sampling-based Watermarking.** For sampling-based watermarking, EXP (Aaronson, 2023) embeds watermarks through exponential minimum sampling at the token level, while Fu et al. (2024); Kuditipudi et al. (2024) build on this method to enhance text diversity further. Zhu et al. (2024) adopts contrastive decoding, whereas SynthID (Dathathri et al., 2024) proposes a tournament sampling scheme that balances high detectability with minimal quality degradation. At the sentence level, SemStamp (Hou et al., 2024a) uses locality-sensitive hashing (LSH) to partition the semantic space into watermarked and non-watermarked regions, while k-SemStamp (Hou et al., 2024b) further refines this process via K-Means clustering.

### A.3  MULTI-BIT LLM WATERMARK

Existing training-free multi-bit watermarking methods can be broadly categorized into two groups:

**Message-enumeration methods.** These approaches hash the message together with its context to partition red/green token lists and, at decoding, enumerate all candidate messages to count green tokens and recover the embedded bits (Fernandez et al., 2023). To reduce complexity, later work embeds portions of the message into separate blocks (Wang et al., 2024; Cohen et al., 2025), but full enumeration is still required within each block. While this yields high decoding accuracy, the exponential growth in candidates severely limits practicality for longer messages.

**Bit-allocation methods.** These methods pseudorandomly assign one or more bit positions to each token and embed them using 1-bit watermarking; decoding simply compares red/green token counts (Yoo et al., 2024). Enhancements include segmenting bits for balanced allocation and adding error-correcting codes to improve robustness (Qu et al., 2025; Chao et al., 2024; Fairoze et al., 2023; Li et al., 2024), dynamically adjusting token lists and using clustering-based decoding (Xu et al., 2025), achieving distortion-free embedding (Zamir, 2024; Boroujeny et al., 2024), and extending unbiased watermarking schemes (Feng et al., 2025; Jiang et al., 2025; Hu et al., 2024; Wu et al., 2023b).

## B  ADDITIONAL RESULTS

Additional experiments under robustness attacks are shown in Figures 7, 8, and 9.

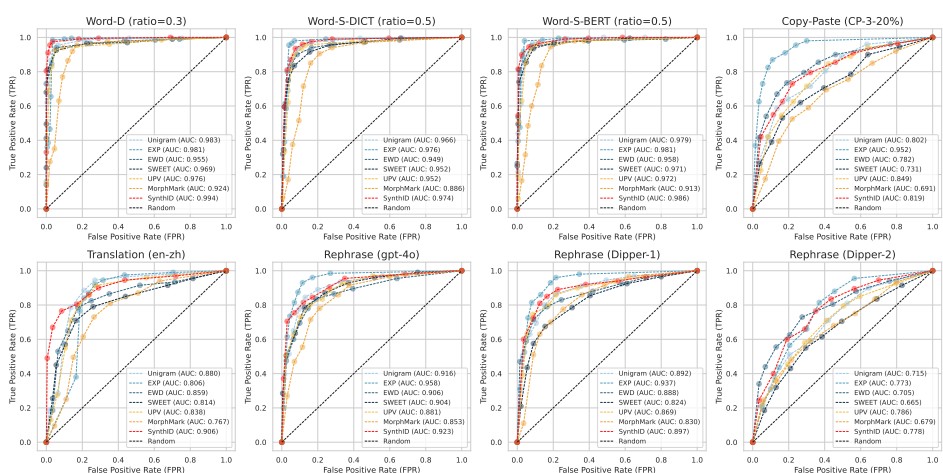

Figure 7: The AUROC curve of Dream-7B-Base watermarked text under various attacks on C4.

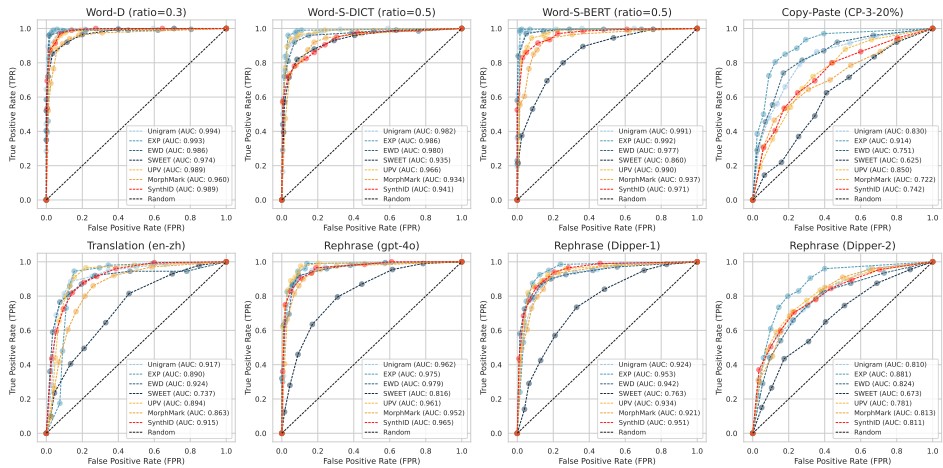

Figure 8: The AUROC of LLaDA-8B-Base watermarked text under various attacks on OpenGen.

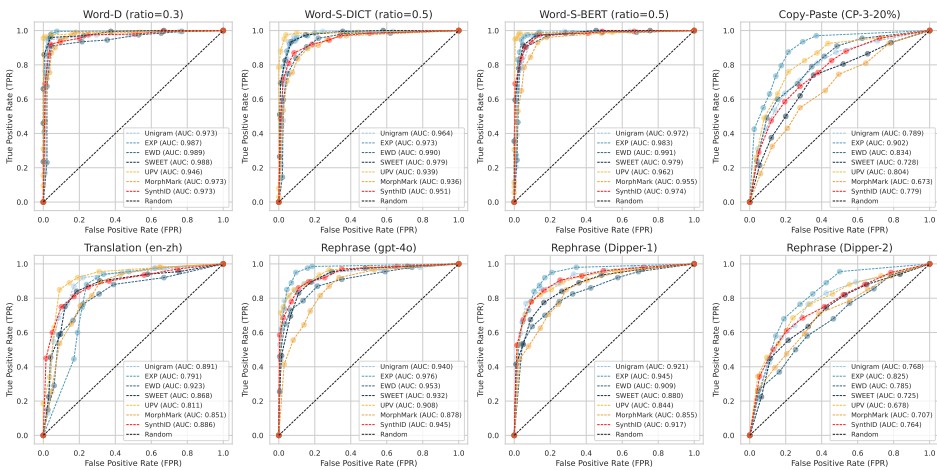

Figure 9: The AUROC of Dream-7B-Base watermarked text under various attacks on OpenGen.

| Model | Watermark | ATTACK TYPE (APPENDIX I) | | | | | | | | |
|-------|-----------|----------|--------|----------|--------|------------|-------------|-----------|-----------|-----------|
| | | No-Attack | Word-D | Word-S-D | Word-S-B | Copy-Paste | Translation | Rephrase-1 | Rephrase-2 | Rephrase-3 |
| LLADA-8B | Ripple (Unigram) | 0.986 | 0.972 | 0.957 | 0.962 | 0.712 | 0.882 | 0.933 | 0.888 | 0.734 |
| | PatternMark (Chen et al., 2025) | 0.962 | 0.959 | 0.945 | 0.953 | 0.699 | 0.843 | 0.902 | 0.814 | 0.672 |
| DREAM-7B | Ripple (Unigram) | 0.992 | 0.983 | 0.966 | 0.979 | 0.802 | 0.880 | 0.916 | 0.892 | 0.715 |
| | PatternMark (Chen et al., 2025) | 0.976 | 0.971 | 0.953 | 0.970 | 0.756 | 0.822 | 0.878 | 0.854 | 0.686 |

Table 3: The AUROC of Ripple and PatternMark watermarked texts on C4 under various attacks.

## C BACKBONE MODELS

In this work, we utilize two popular open-source dLLMs as follows:

### C.1 LLADA FAMILY

LLaDA series model (Nie et al., 2025b) is released in February 2025 and includes two versions: LLaDA-8B-Base and LLaDA-8B-Instruct. Its training objective is to maximize an upper bound on the model's negative log-likelihood, making LLaDA a generative model. This objective naturally enables both in-context learning and instruction-following capabilities while ensuring Fisher consistency, thus supporting scalability to large datasets and models. The pre-training process uses a fixed sequence length of 4,096 tokens and incurs a total computational cost of 0.13 million H800 GPU hours, comparable to autoregressive models of similar scale and dataset size.

### C.2 DREAM FAMILY

Dream series model (Ye et al., 2025) is released in April 2025, with two versions: Dream-v0-Base-7B and Dream-v0-Instruct-7B. It leverages the weights of the autoregressive LLM Qwen 2.5 7B as a non-trivial initialization for the diffusion language model and employs a masked diffusion paradigm. The training dataset covers text, mathematics, and code, mainly sourced from Dolma v1.7, Open-Coder, and DCLM-Baseline, with several pre-processing and curation pipelines. Pretraining was conducted on 96 NVIDIA H800 GPUs over 256 hours, using a mixture of the aforementioned corpora totaling 580 billion tokens.

## D HYPERPARAMETERS ANALYSIS

In this section, we conduct a detailed analysis of several key hyperparameter settings in dLLMs on a demo dataset of 50 samples, focusing particularly on the effects of different diffusion steps, watermark strength, remask ratios, and token length for generation efficiency.

### D.1 DIFFUSION STEPS

Figure 6a illustrates the effect of varying the number of diffusion steps $T$ from 1 to 128 on watermark F1 score and perplexity (PPL) for sequences of 128 tokens. As expected, the F1 score increases with more diffusion steps, while PPL gradually decreases. This is because additional steps facilitate better denoising and more effective watermark injection, improving both text quality and watermark detectability. Considering this trade-off, we set the number of diffusion steps equal to the sequence length, generating one token per step.

### D.2 WATERMARK STRENGTH

As shown in Figure 6b, we further study the impact of different watermark strengths under the Unigram scheme, focusing on their effects on the false positive rate (FPR) and PPL. Results indicate that stronger watermark signals reduce FPR but also degrade text quality. To balance this trade-off, we set the initial watermark strength $\delta$ to 2.0. During watermark calibration, if the watermark score $z_1$ falls below a predefined threshold $\alpha$, we increase $\delta$ to 4.0. Conversely, if the PPL score $z_2$ exceeds the fluency threshold $\beta$, we reduce $\delta$ to 1.5. Note that these watermark strength settings are empirical and may require dynamic adjustment depending on practical scenarios.

---

**Algorithm 2:** Diffusion Large Language Models Watermark Detection

---

**Input:** Target model $\Theta$, suspect text $y$, watermark key $\xi$, watermark algorithm $\mathcal{W}$, threshold $\tau$
**Output:** 1 or 0 (whether the text is watermarked)

1   Select the watermark detector matching the watermarking algorithm $\mathcal{W}$: $\mathsf{Detect}(\cdot) \hookleftarrow \mathcal{W}$
2   Compute the watermark score from the model, text, and key: $\tau' = \mathsf{Detect}(\Theta, y, \xi)$
3   **if** $\tau' \geq \tau$ **then**
4     |   **return** 1 (Watermarked)
5   **else**
6     |   **return** 0 (Unwatermarked)
7   **end**

---

### D.3   REMASK RATIO

In the watermark calibration step, tokens that reach threshold values are remasked. We test different remask ratios, as shown in Figure 6c. The results indicate that varying the remask ratio has a relatively minor impact on final outcomes. This is because high-entropy or high self-information tokens are prioritized for adjustment, as they have a more decisive effect on text quality and detectability. For efficiency, we set the remask ratio to $0.25$.

### D.4   TOKEN LENGTH

To evaluate the efficiency of the Ripple for dLLMs, we record the time required by LLaDA, Dream, and LLaMA3 models to generate watermarked sequences of varying token lengths (1–256), as shown in Figure 6d. With one token generated per diffusion step, LLaDA and Dream take on average 8.7s and 8.2s, respectively, to generate sequences of 256 tokens—slightly longer than the LLaMA3 model (6.1s), representing only a 17% and 14% increase compared with unwatermarked text. We believe that as dLLMs continue to evolve, it will become feasible to generate multiple tokens per step, achieving higher efficiency while maintaining strong detectability and text quality.

## E   WATERMARK DETECTION

The generic watermark detection procedure is outlined in Algorithm 2, where the specific detection functions for different watermarking algorithms are defined as follows:

### E.1   LOGITS-BASED WATERMARK

Representative logits-based watermarking approaches include Unigram (Zhao et al., 2024), SWEET (Lee et al., 2024), EWD (Lu et al., 2024), and MorphMark (Wang et al., 2025b), and their corresponding watermark detection functions are defined as follows.:

$$f_{\mathrm{d}}^{\mathrm{logits}}(x, \xi; \gamma) = 1 - F_B \left( \underbrace{\sum_{t=1}^{\mathrm{len}(x)} f_{\mathrm{hash}}^{\mathrm{logits}}\left(\xi, \gamma, |\mathcal{V}|\right)_{x_t}}_{\text{number of green list tokens in text } x} \right) \tag{5}$$

where $F_B$ is the cumulative distribution function (CDF) for binomially distributed random variable $B \sim \mathrm{Bin}(\mathrm{len}(x), \gamma)$. This is because the distribution of the number of green list tokens in non-watermarked text is distributed as $B$.

### E.2   EXP WATERMARK

The EXP (Aaronson, 2023) watermark detection function is as follows:

$$f_{\mathrm{d}}^{\mathrm{EXP}}(x, \xi) = 1 - F_G \left( \sum_{t=1}^{\mathrm{len}(x)} - \log(1 - f_{\mathrm{hash}}^{\mathrm{EXP}}\left(\xi, |\mathcal{V}|\right)_{x_t}) \right) \tag{6}$$

where $F_G$ is the CDF for gamma distributed random variable $G \sim \text{Gamma}(\text{len}(x), 1)$. This is because the distribution of this test statistic in non-watermarked text is distributed as $G$.

### E.3 SYNTHID WATERMARK

The SynthID (Dathathri et al., 2024) watermark detection function is as follows:

$$f_d^{\text{SynthID}}(x, \xi) = \frac{1}{mT} \sum_{t=1}^{T} \sum_{l=1}^{m} F_g^{-1} \left( \frac{f_{\text{hash}}^{\text{SynthID}}(x, l, r)}{2^{n_{\text{sec}}}} \right) \tag{7}$$

where $F_g^{-1}$ is the generalized inverse distribution function of a g-value distribution with cumulative density function $F_g$, $h$ is the hash function os SynthID, $r \in \mathcal{R} = \{0, 1\}^{n_{\text{sec}}}$ is a random seed and $n_{\text{sec}} \in \mathbb{N}_+$ is a security parameter.

## F DOWNSTREAM TASK DATASAETS

Following Waterbench (Tu et al., 2024), we evaluate our model on the following datasets:

| Category & Source Data | Task | Metric | Language | #data | Len.Input/Answer |
|---|---|---|---|---|---|
| Copen (Peng et al., 2022) | Concept Probing | F1 Score | Language | 200 | 51.52/1.57 |
| ELI5 (Fan et al., 2019) | Long-form QA | Rouge-L | English | 200 | 41.04/236.6 |
| GSM8K (Cobbe et al., 2021) | Math Reasoning | Edit Similarity | English | 200 | 86.25/52.78 |
| CNN/DailyMail (See et al., 2017) | Text Summary | Rouge-L | English | 200 | 883.51/155.4 |

Table 4: A statistical overview of the datasets employed in our downstream tasks experiments.

### F.1 CATEGORY 1 (SHORT INPUT, SHORT OUTPUT)

As the input and answer length decides how much information the watermarking algorithm can hide, we first choose two tasks that have short input and answer length to disturb the watermarking methods. We use the concept probing dataset Copen (Peng et al., 2022), which contains 200 samples from the CIC and CSJ tasks. Due to the brevity of the outputs, we adopt **F1 score** as the evaluation metric. The `max_new_tokens` parameter is set to **16**.

### F.2 CATEGORY 2 (SHORT INPUT, LONG OUTPUT)

To control for the variable of answer length, we choose the Long-form QA task. This category uses 200 samples from the ELI5 (Fan et al., 2019) dataset, a long-form question answering dataset composed of threads from the Reddit forum "Explain Like I'm Five." We use **ROUGE-L** as the evaluation metric. The `max_new_tokens` parameter is set to **128**.

### F.3 CATEGORY 3 (LONG INPUT, SHORT OUTPUT)

To control the variable of input length, we choose the mathematical reasoning tasks. We use 200 samples from the GSM8K dataset (Cobbe et al., 2021), which was created to support multi-step question answering for basic math problems. We evaluate performance using **Edit Similarity**. The `max_new_tokens` parameter is set to **64**.

### F.4 CATEGORY 4 (LONG INPUT, LONG OUTPUT)

To control both input and output length, we choose the text summary task. This category includes 200 samples from the widely used CNN/DailyMail dataset (See et al., 2017), which consists of unique news articles written by journalists and supports both extractive and abstractive summarization. **ROUGE-L** is used as the evaluation metric. The `max_new_tokens` parameter is set to **256**.

# G EVALUATION METRICS

We adopt a suite of metrics to evaluate the watermarking system across three key aspects: detectability, text quality, and robustness.

## G.1 WATERMARK DETECTABILITY

To evaluate watermark detection performance, we define the following standard terms:

- **True Positive (TP)**: A watermarked text is correctly classified as watermarked.
- **False Positive (FP)**: A clean (non-watermarked) text is incorrectly classified as watermarked.
- **True Negative (TN)**: A clean text is correctly classified as non-watermarked.
- **False Negative (FN)**: A watermarked text is incorrectly classified as clean.

Based on these definitions, we compute several derived metrics:

- **True Positive Rate (TPR)** measures the proportion of actual watermarked texts correctly identified:

$$\text{TPR} = \frac{\text{TP}}{\text{TP} + \text{FN}} \tag{8}$$

- **True Negative Rate (TNR)** measures the proportion of unwatermarked texts correctly identified:

$$\text{TNR} = \frac{\text{TN}}{\text{TN} + \text{FP}} \tag{9}$$

- **False Positive Rate (FPR)** measures the fraction of unwatermarked texts mistakenly classified as watermarked:

$$\text{FPR} = \frac{\text{FP}}{\text{FP} + \text{TN}} \tag{10}$$

- **False Negative Rate (FNR)** measures the fraction of watermarked texts mistakenly classified as unwatermarked:

$$\text{FNR} = \frac{\text{FN}}{\text{TP} + \text{FN}} \tag{11}$$

- **F1 Score** A balanced measure that captures both precision and recall:

$$\text{F1} = \frac{2 \cdot \text{TP}}{2 \cdot \text{TP} + \text{FP} + \text{FN}} \tag{12}$$

## G.2 TEXT QUALITY

To evaluate text fluency and downstream task performance, we use the following metrics:

- **Perplexity (PPL)** measures how well a language model predicts the next token. Lower PPL indicates higher fluency:

$$\text{PPL}(x) = \exp\left(-\frac{1}{|x|} \sum_{i=1}^{|x|} \log p(x_i | x_{<i})\right) \tag{13}$$

where $x$ is the generated text and $p(x_i | x_{<i})$ is the conditional probability from a pre-trained language model.

- **ROUGE-L** evaluates the longest common subsequence (LCS) between generated and reference texts. It captures sequence-level similarity and is widely used in text generation evaluation:

$$\text{ROUGE-L} = \frac{\text{LCS}(p, r)}{\text{len}(r)} \tag{14}$$

- **Edit Similarity** measures token-level similarity by computing normalized edit distance:

$$\text{EditSim} = 1 - \frac{\text{EditDistance}(x, x')}{\max(|x|, |x'|)} \tag{15}$$

where $x$ and $x'$ are the generated and reference texts, respectively.

## G.3 ROBUSTNESS

To evaluate how well watermark detection withstands distribution shifts and text perturbations, we measure robustness using the AUC (Area Under the ROC Curve) metric:

- **AUC** evaluates the overall ability of the watermark detection system to distinguish between watermarked (positive) and non-watermarked (negative) texts. It is computed as the area under the ROC curve, which plots the True Positive Rate (TPR) against the False Positive Rate (FPR) across various thresholds. An AUC of 1.0 indicates perfect classification, 0.5 corresponds to random guessing, and values below 0.5 suggest performance worse than random. As a threshold-independent metric, AUC provides a comprehensive measure of the model's discriminative power, particularly useful under varying detection conditions.

# H BASELINE SETTINGS

To ensure consistency and reproducibility, we utilize the MarkLLM toolkit (Pan et al., 2024) to implement both the baseline models and our proposed approach, as outlined below.

- **Unigram** proposed by Zhao et al. (2024), the details of the parameters are as follows: $\gamma = 0.5$, $\delta = 2.0$, $\xi = 15485863$, z_threshold = 4.0

- **EXP** proposed by Aaronson (2023), the details of the parameters are as follows: prefix_length = 0, $\xi = 15485863$, p_value = 1e-4

- **UPV** proposed by Liu et al. (2024a), the details of the parameters are as follows: $\gamma = 0.5$, $\delta = 2.0$, z_threshold=4.0, detect_mode = key, sigma = 0.01, prefix_length = 1

- **SWEET** proposed by Lee et al. (2024), the details of the parameters are as follows: $\gamma = 0.5$, $\delta = 2.0$, prefix_length = 0, z_threshold = 4.0, entropy_threshold = 0.9, $\xi = 15485863$

- **EWD** proposed by Lu et al. (2024), the details of the parameters are as follows: $\gamma = 0.5$, $\delta = 2.0$, $\xi = 15485863$, prefix_length = 0, z_threshold=4.0

- **MorphMark** proposed by Wang et al. (2025b), the details of the parameters are as follows: $\gamma = 0.5$, delta_ewd = 1.25, prefix_length = 0, f_scheme = time, window_scheme = left, $\xi = 15485863$, z_threshold=2.0

- **SynthID** proposed by Dathathri et al. (2024), the details of the parameters are as follows: n = 1, sampling_size = 65536, seed = 0, mode = "non-distortionary", num_leaves = 2, context_size = 0, detector_type = "mean", threshold = 0.52

# I ATTACK SETTINGS

Similar to LLaDA, Figure 7 presents the robustness evaluation results for the Dream-7B-Base model. The detailed descriptions and specific parameter settings for each attack scenario are as follows:

- **Word-D**: Randomly deletes 30% of the words in the watermarked text.

- **Word-S-DICT**: Replaces 50% of the words with their synonyms based on the WordNet dictionary.

- **Word-S-BERT**: Uses BERT embeddings to replace 50% of the words with context-aware synonyms.

- **Copy-Paste**: Retains only 20% of the watermarked text, distributed across three locations within the document.

- **Translation**: Translates the text from English to Chinese and back to English using a fine-tuned T5 translation model[1].

- **Paraphrasing (GPT-4o)**: Uses the GPT-4o API to rewrite the text with low creativity (temperature = 0.2).

---

[1]https://huggingface.co/utrobinmv/

- **Paraphrasing (Dipper-1)**: Applies the DIPPER model to perform paraphrasing with moderate lexical and order diversity while preserving sentence structure (lex_diversity = 40, order_diversity = 40, max_new_tokens = 200, do_sample = True, top_p = 0.75).

- **Paraphrasing (Dipper-2)**: Further increases lexical and order diversity using DIPPER, generating more diverse paraphrases (lex_diversity = 80, order_diversity = 80, max_new_tokens = 200, do_sample = True, top_p = 0.75).

## J  GPT-4 TEMPLATE

The details of the GPT-4 evaluation template used in Figure 5 are as follows:

---

**GPT-4 Judge Template**

"You are given a prompt and a response, and you need to grade the response out of 100 based on: Accuracy (20 points) - correctness and relevance to the prompt; Detail (20 points) - comprehensiveness and depth; Grammar and Typing (30 points) - grammatical and typographical accuracy; Vocabulary (30 points) - appropriateness and richness. Deduct points for shortcomings in each category. Note that you only need to give an overall score, no explanation is required."

---

## K  DISTINGUISHING HUMAN-WRITTEN TEXT

Following (Liang et al., 2023), we evaluated our method on the TOEFL dataset, which consists of non-native English writing samples, as illustrated in Figure 10. The results demonstrate that our approach reliably detects watermarked text on dLLMs (including LLaDA and Dream models), whereas non-native English samples are prone to misclassification by existing AIGT (AI-generated text) detection methods. These findings underscore the practicality and robustness of our watermarking framework Ripple, which achieves an almost zero false positive rate (FPR).

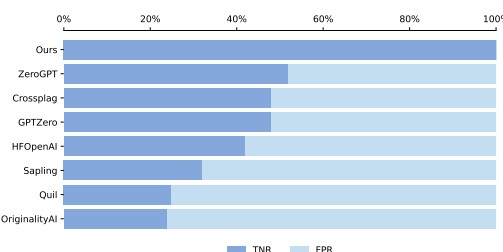

Figure 10: Comparison of AIGT detection methods and Ripple on human-written TOEFL dataset.

## L  REMASK STRATEGIES

Remasking strategies in diffusion sampling are designed to control the generation order of tokens. The various remask strategies used in Figure 5 are detailed as follows:

- **Random**: Tokens are generated in a purely random order, which may lead to unstable performance in certain tasks.

- **Confidence**: Tokens are ranked based on their top-1 probability and generated in descending order of confidence.

- **Margin**: Tokens are ranked by the margin between the top-1 and top-2 probabilities and generated in descending order of this margin.

- **Entropy**: Tokens are ranked by the entropy (Shannon, 1948) of their probability distributions and generated in ascending order of entropy.

## M  MAIN THEORETICAL CONCLUSIONS

By adopting a global hash scheme, watermark embedding in dLLMs at each diffusion step becomes context-independent and position-agnostic. This design removes the main obstacle to dLLMs watermarking analyses because the hashing rule now applies uniformly regardless of generation order. As a result, for watermarking methods based on red–green vocabularies, we can directly build on the Unigram watermark (Zhao et al., 2024) and, by analogy with its theoretical guarantees in LLMs, extend these results to dLLMs, yielding the following conclusions:

### M.1  QUALITY GUARANTEE

Consider $\mathbf{y}_t$ as the input to the dLLM at diffusion step $t$, denoted as $\mathbf{y}_t = [p_0, \mathbf{y}_{t-1}]$. Fix the green list $G$. let $\delta$ represent the watermark strength. For any $\mathbf{y}$, the $\alpha$-th order Renyi-divergence between the watermarked probability distribution $\hat{\mathbf{p}}_t = \hat{\mathbf{p}}_t(\cdot \mid \mathbf{y}_t)$ at time step $t$ and the original probability distribution $\mathbf{p}_t = \mathbf{p}_t(\cdot \mid \mathbf{y}_t)$ satisfies:

$$\forall \boldsymbol{h}, \max\left(D_\alpha\left(\hat{\mathbf{p}}_t \| \mathbf{p}_t\right), D_\alpha\left(\mathbf{p}_t \| \hat{\mathbf{p}}_t\right)\right) \leq \min\left\{\delta, \alpha\delta^2/8\right\}. \tag{16}$$

Equation 16 guarantees that the watermarked distribution $\hat{p}_t$ stays within a $\delta$-controlled bound of the original distribution $p_t$ under virtually all common probability distance measures $D_\alpha$, ensuring minimal distortion of the model's outputs (Dwork et al., 2006; Dong et al., 2020).

### M.2  TYPE I ERROR

Consider $\mathbf{y}$ as any fixed text. Define $C_{\max}(\mathbf{y}) := \max_{i\in[N]} \sum_{j=1}^n \mathbf{1}(y_j = i)$ and $V(\mathbf{y}) := \frac{1}{n}\sum_{i=1}^N (\sum_{j=1}^n \mathbf{1}(y_j = i))^2$. With probability $1 - \alpha$ (over only the randomness of $G$):

$$z_{\boldsymbol{y}} \leq \sqrt{\frac{64V(\boldsymbol{y})\log(9/\alpha)}{1-\gamma}} + \frac{16C_{\max}(\boldsymbol{y})\log(9/\alpha)}{\sqrt{n\gamma(1-\gamma)}}. \tag{17}$$

Equation 17 ensures that, for any non-watermarked text with enough diversity, the watermark detector's z-score stays small so by setting the threshold $\tau$ based on easily computed statistics $V(y)$ and $C_{\max}(y)$, we can guarantee the false-positive rate stays below a chosen level $\alpha$.

### M.3  TYPE II ERROR

Assume "average-high entropy" and "homophilly" to be valid with appropriate parameters, and in addition $n \geq \widetilde{\Omega}(\log(1/\beta)/\delta^2)$, then with probability $1 - \beta$:

$$z_{\boldsymbol{y}} \geq \Omega\left(\left(e^\delta - 1\right)\sqrt{n\gamma(1-\gamma)}\right) \tag{18}$$

To control false negatives (Type II errors), we assume that generated text is sufficiently diverse on average ("average-high entropy") and that increasing the probability of a green-list token does not reduce its likelihood in future steps ("homophily"). Under these conditions and with sufficiently long text, Equation 18 shows that watermarked sequences produce z-scores scaling like $\delta\sqrt{n}$, while non-watermarked text remains $O(1)$. This large margin allows setting a threshold $\tau$ that simultaneously keeps both Type I and Type II errors exponentially small as n grows.

### M.4  SECURITY PROPERTY

Let $y = [y_1, \cdots, y_n]$ represent the watermarked sequence. Suppose the adversary $\mathcal{A}$ with black-box input-output access to the large language diffusion model and outputs a modified text $\mathbf{u} = [u_1, \cdots, u_m]$. Following $z_{\mathbf{y}} = (|\mathbf{y}|_G - \gamma n)/\sqrt{n\gamma(1-\gamma)}$, we calculate z-score $z_{\mathbf{y}}$ and $z_{\mathbf{u}}$. Assume edit distance between $\mathbf{y}$ and $\mathbf{u}$ (denoted as $\eta$) satisfies $\eta < n$. Then we have

$$z_{\boldsymbol{u}} \geq z_{\boldsymbol{y}} - \max\left\{\frac{(1+\gamma/2)\eta}{\sqrt{n}}, \frac{(1-\gamma/2)\eta}{\sqrt{n-\eta}}\right\} \tag{19}$$

In particular, when $\eta \leq \frac{2\gamma n}{(1+\gamma/2)^2}$, we can drop the second term in the max.

Equation 19 shows that the z-score of a watermarked text drops by at most a calculable amount under editing, so for high-entropy sequences and a suitable threshold, our watermark framework remains detectable even after $O(n)$ arbitrary edits, offering strong robustness.

## N  CASE STUDY

Next, we present examples of texts generated using different watermarking algorithms under the Ripple framework. Here, **L** denotes texts generated by LLaDA-8B-Base, **D** denotes texts generated by Dream-7B-Base, **W** denotes watermarked texts, and **U** denotes non-watermarked texts.

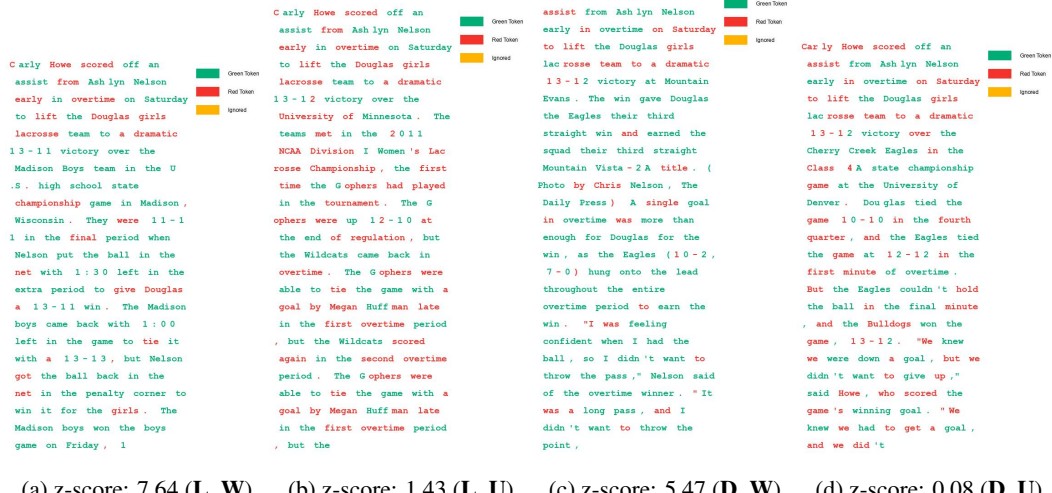

(a) z-score: 7.64 (**L**, **W**)  (b) z-score: 1.43 (**L**, **U**)  (c) z-score: 5.47 (**D**, **W**)  (d) z-score: 0.08 (**D**, **U**)

Figure 11: The Visualization of Unigram Watermarked Text on dLLMs with the Ripple Framework

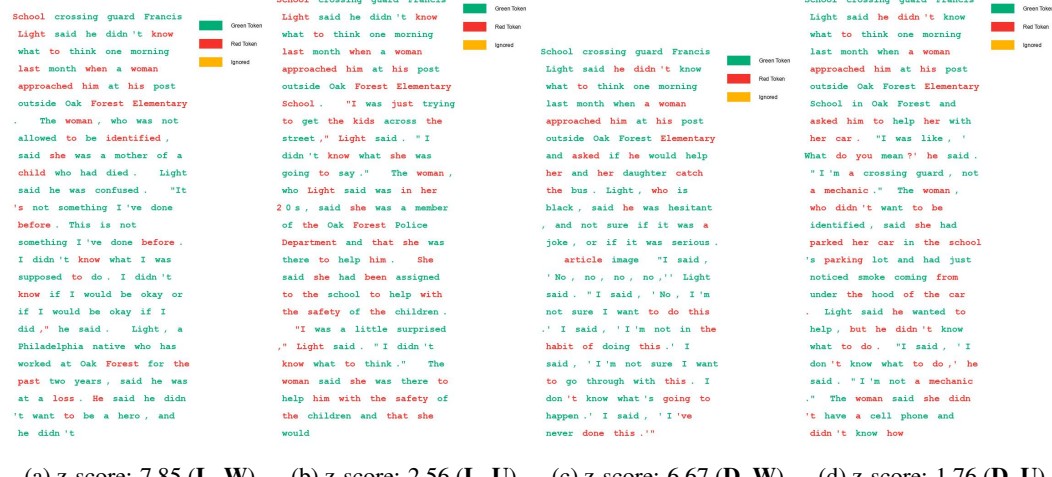

(a) z-score: 7.85 (**L**, **W**)  (b) z-score: 2.56 (**L**, **U**)  (c) z-score: 6.67 (**D**, **W**)  (d) z-score: 1.76 (**D**, **U**)

Figure 12: The Visualization of EWD Watermarked Text on dLLMs with the Ripple Framework

## O   FUTURE WORK

Building on our current study, several avenues remain open for further investigation:

- **Adaptive watermarking strategies.** Develop mechanisms that adjust watermark strength or placement dynamically based on context to improve robustness and maintain text quality.
- **Security under adversarial settings.** Analyze and enhance the resilience of dLLM watermarks against deliberate attacks such as removal, evasion, stealing, or model fine-tuning.
- **Cross-modal watermarking.** Extend watermarking methods to multimodal diffusion models, exploring how textual, visual, and audio modalities can be jointly watermarked.
- **Watermark–utility trade-offs.** Systematically study the balance between watermark strength, detection accuracy, and the utility/performance of the underlying model.
- **Multi-bit watermarking schemes.** Explore higher-capacity schemes beyond single-bit watermarking, leveraging the block-wise generation property of dLLMs.

## P   LLM USAGE

We used large language models (LLMs) solely as general-purpose assistive tools for language polishing and grammar improvement of the manuscript. LLMs did not contribute to research ideation, experimental design, data analysis, or substantive writing, and therefore are not considered contributors or co-authors of this work. The authors take full responsibility for the accuracy and integrity of all content in this paper, including any text revised with the assistance of LLMs.

