# OpenReview forum: "Signed in Ink, Hidden in Noise: Watermarking Diffusion Large Language Models"
_ICLR.cc/2026/Conference — Submitted to ICLR 2026_

### Official Review · Reviewer_7BVB · 2025-10-31

**Soundness:** 2
**Presentation:** 3
**Contribution:** 2
**Rating:** 2
**Confidence:** 4

**Summary:**

The paper is the first to study watermarking in diffusion language models. The proposed framework, Ripple, consists of two main steps:

1. Watermark Injection: Inject the watermark only into the tokens that are to be unmasked in each diffusion step.
2. Watermark Calibration: After all tokens are sampled, the framework performs calibration by computing the contribution of each token to the overall watermark signal. Tokens that contribute less to the watermark signal are resampled with increased watermark strength. In addition, the framework evaluates how each token contributes to the overall perplexity (PPL) by computing the PPL while leaving each token out, and then reduces the watermark strength for tokens that contribute more to the PPL.

Experiments are conducted on LLaDa-8B and Dream-7B.

**Strengths:**

1. Studying watermarking for diffusion language models is a valuable research direction.
2. The presentation is mostly clear, and the figures are illustrative.

**Weaknesses:**

w1: Lack of clarity in identifying the limitations of adapting autoregressive watermarking to diffusion language models.

The paper does not clearly articulate the limitations of extending watermarking methods designed for autoregressive models to diffusion language models. As a result, the specific benefits of the proposed algorithm relative to existing watermarking techniques remain unclear.

* The authors claim that “after watermark tokens are generated at each step, a large portion of them are remasked, causing the watermark signal to dissipate.”
  It is unclear what is meant by a “large portion” being remasked. In each diffusion step, as introduced in Section 2.1, only ( $L/T$ ) tokens are selected from the masked token set ( $\mathcal{M}$ ) and added to the unmasked set ( $\mathcal{U}$ ), without reversal. Therefore, if the chosen tokens in a given diffusion step are watermarked, it is not evident why remasking would cause the watermark signal to dissipate.

* The issue of degraded text quality is not sufficiently discussed. The main limitation (I think) appears to be that the remasking strategy depends on the probability of the next-token prediction (e.g., Confidence or Margin strategies in Appendix L). Consequently, injecting the watermark before remasking may alter which tokens are unmasked, thereby impacting text quality. If this is indeed the case, the authors should have included the remasking strategy in the main paper and discussed how watermarking influences the remasking process.

* Even acknowledging this limitation, it remains unclear how the proposed algorithm addresses it. Because token probability distributions are interdependent, watermarking earlier tokens inherently affects the probabilities of subsequent tokens. From this design perspective, the proposed approach (without calibration) does not convincingly demonstrate an improvement in text quality.

* The evaluation baselines are also insufficient. Watermarks whose hash values are independent of context should be easily adaptable to diffusion language models. However, the paper only compares an autoregressive watermarked model with those existing watermarks vs diffusion language models with Ripple (on top of existing watermarks) for diffusion language models (in Table 2). Since multiple factors vary at the same time, it is difficult to isolate the impact of watermarking only the unmasked tokens, which appears to be the main distinguishing feature of the proposed framework.
---

w2. The proposed algorithm lacks strong design motivation

Prior to the calibration step, the primary modification introduced by the framework is identifying which positions to apply the watermark and restricting watermarking to those positions. While this may be a reasonable design choice, I am not convinced that it constitutes a substantial research contribution; rather, it represents a practical refinement.

#### **Issues with the Calibration Process**

* The claim that increasing the watermark strength for tokens that contribute marginally to the watermark signal would lead to a degradation in the PPL performance, as demonstrated in many prior works. There seems to be an inherent trade-off between PPL and watermark detectability.

* To improve text quality, the framework computes the perplexity (PPL), which would significantly increase computational cost and reduce efficiency, while also rendering comparisons with other watermarking methods less fair (as prior watermarks do not use the PPL to refine the text). Furthermore, when the PPL is high, the proposed algorithm computes *self-information*, equivalent to leave-one-out PPL for each token. This process dramatically increases computational overhead and inefficiency. In the experimental setup, the worst-case scenario involves computing PPL ( $(128 + 1) \times 128 = 16,512$) times, which is computationally prohibitive.

* More importantly, the notion of self-information does not align with the authors' claim that it reflects “how much a token contributes to the overall perplexity of the sentence.” Removing a token disrupts sentence smoothness and grammaticality, potentially altering the entire sentence structure. Given that PPL is the exponential of the loss, a more appropriate and straightforward approach would be to analyze each token’s individual loss directly, which accurately quantifies its contribution to overall perplexity.

* The claimed advantage of “avoiding left-to-right regeneration” primarily results from the use of a global hash function, rather than from any unique property of the proposed algorithm’s design.

---

Overall, while the paper addresses an interesting and timely problem, it falls short of identifying and addressing the fundamental limitations of adapting autoregressive watermarking techniques to diffusion language models, both conceptually and empirically. As a result, the contribution of the work remains questionable. The algorithmic design lacks sufficient novelty and clarity, and the evaluation setup does not convincingly substantiate the claimed advantages.

**Questions:**

Please see my comments on weakness. Here are some minor comments:

* The function $s$ should also take the watermark key ( $\xi$ ) as input, since the watermark key is required for detection in Section 3.4.
* $z_2$ is not defined in the paper but appears in Algorithm 1. Its meaning should be clarified before being referenced in Section 3.3.
* In Table 1, the authors should report TPR@1%FPR to provide a clearer view of the trade-off between detection rate and false alarm rate.
* In Figure 6(d), the runtime should be reported both with and without calibration, and with and without token selection prior to watermarking, under the same model configuration.
* In Table 2, the meaning of FPR should be explicitly stated. Moreover, the FPR values are high for all settings on the T1 task. If the FPR is this high, it is questionable whether the results from this task provide any meaningful insight into the effectiveness of watermarking on downstream tasks.

---

### Official Review · Reviewer_WZ51 · 2025-10-31

**Soundness:** 2
**Presentation:** 1
**Contribution:** 2
**Rating:** 2
**Confidence:** 4

**Summary:**

This paper introduces Ripple , a systematic watermarking framework specifically designed for Diffusion Large Language Models (dLLMs). The method operates in two stages: a "watermark injection" phase that progressively embeds a signal using a global hash scheme, and a "watermark calibration" phase. This calibration stage functions as a reject-resampling process , assessing the final output for both watermark strength and text quality (e.g., PPL) and selectively regenerating tokens that fail to meet predefined thresholds.

**Strengths:**

1.  Thorough Experimentation: The paper provides a comprehensive and detailed set of experiments, evaluating the proposed method under various conditions and against relevant baselines.
2.  Clarity of Method: The proposed approach is well-explained, logical, and straightforward for the reader to understand.

**Weaknesses:**

See Questions below.

**Questions:**

1.  **Significant Lack of Novelty:** My primary concern is the paper's limited novelty, as the proposed method appears to be a straightforward composition of two pre-existing techniques.
    * The first component, the "global hashed LLM watermark," seems conceptually identical to the method already thoroughly discussed in Unigram. The theoretical justification in Appendix M appears to be a direct reuse of the conclusions from Unigram paper, without offering any original proofs or new theoretical insights adapted to this work.
    * The second component, the "reject-resampling" procedure used to ensure quality and strength, has already been deeply investigated and validated in WaterMax [1]. The new contribution in this paper is minor from my perspective.

2.  **Insufficient Discussion of Computational Overhead:** The calibration (reject-resampling) step is known to be the main computational bottleneck for this entire class of watermarking methods. The paper glosses over this critical issue with a single sentence: "While calibration slightly lowers efficiency, it enhances watermark detectability, and future dLLMs may alleviate this overhead by generating multiple tokens per step." This is highly insufficient. A practical method must include a rigorous analysis of the latency and computational cost introduced by this step.

3.  **Lack of Clarity in the Calibration Process:** The description of the calibration mechanism is overly simplistic in Section 3.3 and leaves several key operational details unanswered.

    * The paper states, "we remask the tokens that contribute minimally to the watermark signal, which are assigned low confidence by the detector." (a) How is a "low confidence token" formally defined and quantified? (b) What is the criterion for remasking (e.g., what percentage of tokens are remasked, or what is the "low confidence" threshold)? This appears to be a critical, unstated hyperparameter.

    * The paper then states, "tokens with higher entropy are prioritized." (c) How is this "prioritization" scheme practically implemented? (d) What is the relationship between this "high entropy" prioritization and the "low confidence" criterion mentioned earlier? Which one takes precedence, or how are they combined?

    * (e) How is the self-information metric (Eq. 4) practically computed using the dLLM?

4.  **Missing Ablation Studies:** As highlighted in the previous point, the calibration process introduces numerous crucial hyperparameters (e.g., the definition of "low confidence," the remasking ratio, the prioritization scheme). However, the main body of the paper provides no ablation studies to analyze the method's sensitivity to these parameters. This makes it difficult to assess the robustness of the reported results and understand the practical trade-offs involved in tuning the method.


[1]WaterMax: breaking the LLM watermark detectability-robustness-quality trade-off

---

### Official Review · Reviewer_FGw4 · 2025-10-31

**Soundness:** 3
**Presentation:** 3
**Contribution:** 3
**Rating:** 6
**Confidence:** 5

**Summary:**

In this paper, the authors propose to convert text watermarking methods designed for auto-regressive models to methods working for diffusion LLMs. The stated goal is to propose a comprehensive, functioning baseline against which future work can compare.

The paper identifies two main questions for the conversion:
	1. At what location and time the watermark should be injected during the diffusion process -- e.g. which logit distribution should be modified at each step?
	2. Can we leverage the *non-causality* of the process to better select the positions to watermark provided that maximize quality AND detectability

The authors answer the first question using ad-hoc remask strategies (four simple masking/ordering method described in the appendix).

The second question is answered using a refinement mechanism based on entropy (for detectability) and self-information (for perplexity which is linked to quality). Tokens are remasked and regenerated if the watermarked sequence don't match predifined scores on each metric.

Finally, the authors provide a (very!) comprehensive experimental section using a complete suite of classic text watermarking algorithms and converting them to diffusion LLMs. The evaluation is performed both on standard completion tasks (C4 and OpenGen) as well as downstream tasks (Q&A, maths, summarization and factual questions).

**Strengths:**

**Quality and significance** : The paper has one goal: serve as a baseline for future work on text watermarking for dLLM. It leverages previous art and does some simple yet effective accommodation to make it function in this emerging setup. It correctly identifies key take-aways from auto-regressive models, namely the focus on perplexity as a proxy measuring the deviation from the original text distribution and entropy as the main limiting factor of watermarking strength.

**Exhaustive**: The authors provide an adaptation for a wide array of classical watermarking schemes and evaluate them on multiple dLLM models (LLaDA, Dream) across diverse datasets. The inclusion of both perplexity/detectability benchmarks and downstream task performance  provides a well-rounded and valuable view of the trade-offs.


**Originality**: The paper has no claim to originality, since it is a simple translation of previous art to a new setting. Yet the authors do not oversell the approach as something new, leading me to believe lack of originality should not be considered as a weakness of the paper.

**Weaknesses:**

**Lack of regard with respect to security**: The authors chose to use a global hashing scheme as in Unigram. First, I note that the hashing method is never formally defined (neither in the main body nor in the appendix). Second, though the global hashing method provides the best robustness over all hashing window, it is also know to be highly insecure, since the secret key can be inferred from the LLM output extremely simply  (the hash only depends on the token ID ) -- see [1]. The authors don't mention this important caveat and  how it applies to their method.

**Unclear metrics** : The authors provide both the FPR and TNR in Table 1, found by setting a z score threshold to $4.0$. The reason for this choice is unclear. Even stranger, if I dig into appendix H, I can see that the threshold vary depending on the scheme used... The interpretation also varies: the authors seem to use a $p$-value for EXP, which should **guarantee** a FPR of $10^{-4}$, whereas threshold for a z score only provides guarantees asymptotically (and not even at the same value of $10^{-4}$ !). These choices should be made clear and standardized in the paper. Notably, I would foregeo separating the TPR and FPR altogeter and work at a fixed FPR when possible. Since a sound $p$-value can be derived for most watermarking schemes -- at least Unigram, KGW and EXP, see [2] -- this should not be a problem.    The "DROP" metric in Table 2 is  is never formally defined and i have trouble understanding what it means, is it the drop in the generation metric evaluating the downstream task? In which case, they should be defined in the paper for replicability.

**Questions:**

**Questions**:

1. The choice of the the global hashing is a security vulnerability. This not a problem for KGW or EXP since they can use other hashing techniques and larger windows. The authors however say their framework is dependent on this hashing technique. This should be discussed and outlined in the paper, especially i no alternative can be designed.

2. Please standardize the metric computations using $p$-values when possible, allowing a guaranteed FPR.

3. Please explicitly define the "DROP" metric used in Table 2 in the main body or caption, as its meaning is currently ambiguous.


**Recommendation**: This paper falls in the category of "useful but not ground-breaking". As a researcher in the field, I welcome such a paper as providing a clean, replicable baseline on which future works can be measured against -- i.e. "can we do better than methods formerly designed for auto-regressive models ?". The onus falls on ICLR to know if such work has its place in such a highly competitive environment -- maybe TMLR or other venues which specialize in non-SOTA yet technically correct and useful work would be a better match for such a paper.
With that said, my current recommendation is a *borderline accept*, which, provided my gripes about security and the choice of sound statistical metrics are addressed, I would gladly raise to an *accept*. I don't think however such a paper is *strong accept* material due to its lack of novelty and originality.

**References**:

    - [1] Jovanović, Nikola, Robin Staab, and Martin Vechev. "Watermark stealing in large language models." arXiv preprint arXiv:2402.19361 (2024).

    - [2] Fernandez, Pierre, et al. "Three bricks to consolidate watermarks for large language models." 2023 IEEE international workshop on information forensics and security (WIFS). IEEE, 2023.

---

### Official Review · Reviewer_joK4 · 2025-11-11

**Soundness:** 2
**Presentation:** 3
**Contribution:** 3
**Rating:** 4
**Confidence:** 3

**Summary:**

This paper studies how to watermark diffusion large language models (dLLMs) that generate text via iterative denoising rather than left-to-right token prediction. It proposes Ripple, the first general watermarking framework tailored to this setting. The goal is to embed imperceptible signals into dLLM outputs so that AI-generated text can later be reliably detected while preserving text quality.

**Strengths:**

1.	Novel application: The paper tackles the emerging problem of watermarking diffusion-based text generation models, which have received little prior attention compared to autoregressive LLMs.
2.	General design: The proposed Ripple framework is elegant and model-agnostic, enabling the adaptation of various existing watermarking algorithms (e.g., EXP, SynthID) to the diffusion setting with minimal modification.
3.	Extensive empirical results: Extensive experiments across multiple dLLMs and watermarking methods demonstrate that Ripple achieves high F1 scores and maintains low perplexity, with robustness against paraphrasing and editing attacks.

**Weaknesses:**

1.	This paper proposes a framework that can adapt any logit-based or sampling-based watermarking methods to dLLMs, but not a new watermarking algorithm specifically designed for dLLMs.
2.	In vanilla dLLMs, some of the already-filled tokens might be remasked (to add noise again) depending on the schedule. Under the Ripple framework, the watermarked tokens will not be remasked again. I wonder if this will cause issues in the diffusion process.
3.	The paper claims to be the first to watermark diffusion-based language models. While plausible, this claim may be overstated given concurrent works. The authors could clarify their positioning and emphasize the framework’s generality and robustness rather than chronological priority.

**Questions:**

1.	How does the calibration module exactly work? It is only briefly mentioned in pseudo-codes. Moreover, what is the exact implementation of the global hash scheme?
2.	It is unclear why more diffusion steps improve F1. It seems that no matter how many diffusion steps there are, eventually all tokens will be injected with watermarks.

I am open to adjusting the score if the authors address these concerns.

---

### Meta-Review · Area_Chair_9S9X · 2026-01-05

**Summary:**

The paper proposes a watermarking approach for the diffusion large language models (dLLMs). The introduced Ripple method operates in two stages of watermark embedding and calibration.

Overall, the authors did not engage with the reviewers into the discussion. The initial assessment from the reviewers is to reject the paper.

**Reviewer Concerns:**

Reviewer joK4 is concerned with the positioning of the paper as the first watermarking method for dLLMs. The reviewer suggest that the authors should rather focus on framework’s generality and robustness.

Reviewer FGw4 points out the the hashing method is not defined and can be highly insecure. Additionally, the choice of the metrics is unclear.

Reviewer WZ51 indicates a significant lack of novelty since the proposed method is a composition of the global hashed LLM watermark and reject-resampling procedure from WaterMax. The submission lacks a rigorous analysis of the latency and computational cost introduced by the reject-resampling step. There is no analysis of the methods sensitivity to its parameters (e.g., the remasking ratio or the prioritization scheme).

Reviewer 7BVB argues that the issue of degraded text quality is not sufficiently discussed. Moreover, the evaluation baselines are insufficient. Overall, for this reviewer, the contribution of the work remains questionable. The algorithmic design lacks sufficient novelty and clarity, and the evaluation setup does not convincingly substantiate the claimed advantages.

Taking into account all the above reviews and the lack of responses from the authors, I recommend rejection of this submission.

**Reviewer Scores:**

Reviewer joK4: Score 4 / Confidence 3

Reviewer FGw4: Score 6 / Confidence 5

Reviewer WZ51: Score 2 / Confidence 4

Reviewer 7BVB: Score 2 / Confidence 4

---

### Decision · Program_Chairs · 2026-01-26

Reject